# Factors associated with intrauterine contraceptive device use among women of reproductive age group in Addis Ababa, Ethiopia: A case control study

**Nebiyu Dereje** [1]*, **Biruk Engida** [2], **Roger P. Holland** [2]

1 Department of Public Health, Myungsung Medical College/Myungsung Christian Medical Center, Addis Ababa, Ethiopia, 2 Department of Medicine, Myungsung Medical College/Myungsung Christian Medical Center, Addis Ababa, Ethiopia

☯ These authors contributed equally to this work.

* nebiyudereje2019@gmail.com

**Data Availability Statement:** All relevant data are within the paper and its Supporting Information files.

## Abstract

### Background

Intrauterine Contraceptive Device (IUCD) is a highly effective and reversible modern contraceptive, which is still significantly underutilized in Ethiopia. The aim of this study was to identify factors affecting the use of IUCDs among women of reproductive age group in Addis Ababa.

### Methods

Facility-based, unmatched case-control study was employed among randomly selected cases and controls in selected health centers in Addis Ababa from August to October 2017. The cases (n = 128) were women of reproductive age group who were IUCD users and controls (n = 256) were women of reproductive age group who were users of oral or injectable contraceptives during the study period. After randomly selecting two health center from each sub-city the number of cases and controls were equally allocated to each of the selected health centers. In each selected health center, all eligible cases were enrolled consecutively until the sample size was achieved. Two consecutive controls were selected for each case. Data was collected face-to-face by trained nurses by using structured questionnaire. Factors associated with IUCD use were identified by multi-variable binary logistic regression models using the backward conditional stepwise method.

### Results

In the multi-variable analysis, IUCD use was strongly associated with husbands/partners being supportive of IUCD use (Adjusted OR = 13.24, 95% CI; 5.30–33.02), being literate women (Adjusted OR = 5.31, 95% CI; 1.05–26.93), women having a perception of IUCD does not cause infection (Adjusted OR = 4.38, 95% CI; 1.45–13.26) and the source of information about IUCD being mass-media (Adjusted OR = 3.81, 95% CI; 1.49–9.74).

**Funding:** The author(s) received no specific funding for this work.

**Competing interests:** The authors have declared that no competing interests exist.

## Conclusions

The findings of the study reinforce the need of husbands/partners involvement in the interventions to enhance utilization of IUCD. Moreover, due attention should also be provided for delivering IUCD-related messages in the public mass-media.

## Introduction

Contraceptives reduce the number of abortions, prevent unintended pregnancies, and lower the incidence of death and disability related to complications of pregnancy and childbirth. It is reported that an additional 24 million abortions, 6 million miscarriages, 70,000 maternal deaths and 500,000 infant deaths would be prevented, if all women with an unmet need for contraceptives were able to use modern methods worldwide [1, 2]. However, only 33% of women in Africa used contraceptive methods in 2015 [2]. In Ethiopia contraceptive prevalence rate among married women has shown an increase from 28.6 percent in 2011 to 36 percent in 2016, resulting in a decline of total fertility rate from 4.8 in 2011 to 4.6in 2016 [3]. Nevertheless, Ethiopia still remains one of the countries with low contraceptive use rate, high fertility rate (the second populous and one of the highest fertility rates in Africa, with about 40% of its population is under fifteen) and high number of unintended pregnancies [4–8]. Moreover, the maternal mortality ratio is still high (412/100,000 women) [3]. Therefore, increasing the uptake of modern contraception is one of the solutions in order to decrease the fertility rate and to reduce the unmet need for family planning [4, 6, 9, 10].

The IUCD) is one of the most reliable and the most widely used reversible modern contraceptive method worldwide and has the potential to reduce the overall number of unintended pregnancies more than any other contraceptive method [11, 12]. Despite the fact that the copper-bearing IUCD brand TCu-380A is widely available in Ethiopia and is provided free of charge in government healthcare facilities, it is still very much underutilized [3]. Unlike other countries where IUCD is widely used such as: Scandinavian countries (18%), Asian nations (13%), the Near East and North Africa (12%) [13] and Kenya (4.8%) [13, 14], IUCD use in Ethiopia was only 2%, as reported by the Ethiopian Demographic and Health Survey of 2016 [3].

Even though, consistency and conclusiveness has not yet reached, studies in different parts of Ethiopia have shown several different factors affecting utilization of IUCDs. These factors include: age of the women, her religion, her level of education, a history of abortion, and perceived myths relating to the use [4–7, 9, 10, 15–21]. However, the problem of underuse of IUCD is still prevalent and much study is required to identify locally plausible determinants. Therefore, it is of a great importance to undergo a research in different areas of the city to maximize the scope of the study across a widely different socio-demographic and geographic area and to more precisely identify the current factors that are associated with low IUCD use.

Thus, understanding these factors may help for developing a strategy that can help in improving the utilization of IUCD in Addis Ababa and there by contribute towards decreasing the number of the unintended pregnancies and the fertility rate.

## Materials and methods

### Study setting and design

A facility-based unmatched case control study was conducted from August to October 2017 among women of reproductive age group (15–49 years old) who were seeking family planning

services at the randomly selected health centers in Addis Ababa. Addis Ababa is the capital city of Ethiopia, with the population size of 3,384,569 million people out of which 52.6% were female, according to the 2007 census report [22]. The city is organized in ten administrative sub-cities. A health center in Ethiopia is the lowest level of an officially recognized healthcare facility that can insert IUCD. It is much closer to the community, serves about 25,000 people of the area and provides basic health care services such as: outpatient care, antenatal and delivery services, family planning services, HIV screening and care services, immunization, child health services, and some minor trauma care (e.g., burn dressing changes) [23]. All the health centers provide family planning services regularly, including the insertion and removal of IUCD.

### Study population

The cases were women of reproductive age group (15–49 years old), permanent residents of Addis Ababa (i.e., were residents for ≥ 6 months), and were new users (first time users) of IUCD in the selected health centers during the study period. Whereas, the controls were women of reproductive age group (15–49 years old), permanent residents of Addis Ababa (i.e., were residents for ≥ 6 months), who were new users of hormonal contraceptives (oral, injectable or implants) in the selected health centers during the study period.

### Sampling procedure

The sample size was determined using two population proportion formula by using age as a main determinant factor in IUCD utilization [4]; where 81.5% of IUCD users and 67.6% of non-IUCD users were reproductive women above the age of 25 and considering 95% CI, 80% power of the study, and a one case to two control ratio. The sample size calculated from the above assumptions gave 128 cases and 256 controls.

There were 98 health centers in Addis Ababa, out of which, 20 health centers (two from each sub-city) were selected by employing simple random sampling technique (lottery method). Then, the number of cases and controls were equally allocated to each of the selected health centers. In each health center, all the eligible reproductive age group women who were new IUCD-users were included in the study consecutively until the sample size was achieved.

For each case, two adjacent controls that used OCPs, implants or injectable contraceptives during the study period were selected and if the selected control did not fulfill the inclusion criteria, the next woman who used OCPs, implants or injectable contraceptives was considered.

### Ethical considerations

Ethical clearance was obtained from Myungsung Medical College Institutional Review Board (IRB) and the letter of permission was obtained from Addis Ababa City Regional Health Bureau (S1 File). Subsequently, from each health center an official letter of permission was obtained. Thereafter, informed verbal consent was obtained from each study subject before participation in the interview. Since this study was solely based on interviewing the clients and no invasive procedures was used, informed verbal consent would suffice. The use of verbal consent was approved by the IRB. Moreover, informed verbal consent was more preferable than informed written consent by the study participants. This is because clients in Ethiopia, where the level of literacy is much lower, have a perception that putting a signature might have some legal consequences. Furthermore, signing a paper might hinder the clients from explaining their true feelings and might incline towards responding of only positive responses for the interview questions.

Documentation or recording of the informed verbal consent was possible by reading the information sheet for the clients and requesting for the consent of the participation (S1 Appendix). Whenever the client refuse to participate in the study, her questionnaire was closed and marked as "not consented" and the next family planning client was considered.

## Data collection procedure

A face-to-face interview was conducted immediately after the method of choice of contraception is administered to a woman. Interview was administered by trained nurses and public health professionals by using structured questionnaire. The interviewers were not involved in the family planning service provision. The questionnaire was developed based on previously done similar studies obtained from literature review which were published under a CC-BY license [5, 7, 16, 17, 19]. The structured questionnaire (S1 Appendix) was first prepared in English language and translated into Amharic, the national language of Ethiopia.

Training was given to the data collectors on the purpose of the study, the contents of the questionnaire and on issues related to the confidentiality of the responses and the rights of respondents. A pre-test on 5% of sample size on a health center out of the selected heath centers was done and the questionnaire was corrected and modified based on the pre-test results. Supervision of the data collection procedures was made by the researchers on a daily basis. During these supervisory activities, quality and completeness of gathered information were checked and timely corrections were made to improve the quality, consistency and completeness of data of subsequent interviews.

## Data management and analysis

Modern contraceptives in this study were defined as a product or medical procedure that interferes with reproduction from acts of sexual intercourse [24]. These includes: Sterilization (male and female), IUCDs, Subdermal implants, Oral contraceptives, Condoms (male and female), Injectables, Emergency contraceptive pills, Patches, Diaphragms and cervical caps, Spermicidal agents (gels, foams, creams, suppositories, etc.), Vaginal rings and Sponge. The natural methods of contraceptives were considered as non-modern contraceptives. These includes: Calendar rhythm methods, Withdrawal, Lactational amenorrhea, Abstinence [24].

IUCD users in this study were those women of reproductive age group who have got IUCD inserted for the first time only during the time of study. Whereas, non-IUCD users were those women of reproductive age group who only used hormonal contraceptives (e.g., OCPs, implants or injectable) during the time of study. A woman was considered knowledgeable of the IUCD based on whether she has mentioned IUCD when asked to enumerate the family planning methods she knew during the study. If the woman had mentioned IUCD as one of the family planning methods, she was considered as having knowledge of IUCD as method of contraception.

Data was entered, cleaned and analyzed using SPSS (Statistical Package for Social Sciences) version 20 software. Data were checked for inconsistencies and missing values; then variables were defined, categorized and recoded for analysis. Descriptive summary was explained by frequency and percentage. Factors associated with IUCD use were identified first by bivariate analysis and then multi-variable binary logistic regression analysis using backward conditional stepwise method. Those variables with $p$-value $<0.25$ in the bivariate analysis were included in the multi-variable analysis. Odds ratios (OR), 95% confidence interval (CI) and $p$-value set less than 0.05 were used to determine the statistical significance of the associations between independent variables and the dependent variable. Multi-collinearity between variables was assessed using the multi-collinearity diagnostics (Variance Inflation Factor and tolerance test).

The final multi-variable binary logistic regression model was found to be fit based on the finding of the Hosmer-Lemeshow goodness-of-fit test.

## Results

### Socio-demographic characteristics

A total of 128 cases and 256 controls were interviewed and their data were analyzed. Four (3.1%) of the cases and 43 (16.8%) of the controls were younger than 20 years. Majority of the cases (93.8%) and controls (75.2%) were married. More than three-fourths of the cases (79.6%) and controls (84.4%) were Christian. Only 7 (5.5%) of cases and 34 (13.2%) of controls were illiterate (not able to read and write). About half (46.9%) of cases and 118 (45.7%) of controls were house wives. In the bivariate analysis, IUCD use was associated with age, marital status and educational status of the women (Table 1).

### Characteristics of reproductive history

The majority of cases (94%) and controls (69.4%) had a history of pregnancy and the majority (69.3%) of the pregnancies was planned. About forty percent of cases and thirty percent of controls had three or more births. In a majority of cases (70.8%) and controls (43%), age of their youngest child was 2 years or higher. More than 20% of the cases and controls had history of abortion and more than half of the cases and more than three-fourths of the controls had future plan of giving birth.

Reproductive characteristics of respondents that were associated with IUCD use in the bivariate analysis include; ever being pregnant, age of youngest child, and future plan of fertility (Table 2).

### Contraceptive knowledge and attitude

All of the study participants heard about at least one of the modern family planning methods and majority of the cases (45.3%) and controls (32.6%) knew four or more modern contraceptive methods. As indicated in Table 3 below, about 39% of cases and 80% of controls reported that they received information about family planning from healthcare providers. Almost all of the cases (98%) and less than half of the controls (42.6%) were knowledgeable about IUCD. A

**Table 1. Socio-demographic characteristics of women and its association with IUCD use in Addis Ababa, 2017.**

| Variables | Case (n = 128) | Control (n = 256) | *p*-value |
|---|---|---|---|
| **Age** | | | |
| 15–20 | 43 (89.6%) | 4 (10.4%) | |
| 21–30 | 159 (72.3%) | 61 (27.7%) | 0.001 |
| 31–40 | 44 (44.0%) | 56 (56.0%) | |
| 41–49 | 10 (58.8%) | 7 (41.2%) | |
| **Marital status** | | | |
| Not Married or divorced/widowed | 8 (11.1%) | 64 (88.9%) | |
| Married | 120 (38.5%) | 192 (61.5%) | 0.0001 |
| **Religion** | | | |
| Muslim | 102 (72.9%) | 38 (27.1%) | |
| Christian | 26 (10.7%) | 218 (89.3%) | 0.18 |
| **Educational status** | | | |
| Illiterate | 7 (17.1%) | 34 (82.9%) | |
| Literate | 121 (35.3%) | 222 (64.7%) | 0.024 |

**Table 2. Characteristics of reproductive history and its association with IUCD use among reproductive age women in Addis Ababa, 2017.**

| Variables | Case (n = 128) | Control (n = 256) | *P*-value |
|---|---|---|---|
| **Ever pregnant** | | | |
| No | 8 (9.2%) | 79 (90.8%) | |
| Yes | 120 (40.4%) | 177 (59.6%) | 0.0001 |
| **Number of live birth (parity)** | | | |
| ≤2 births/children | 75 (27.3%) | 200 (72.7%) | |
| >2 births/children | 53 (48.6%) | 56 (51.4%) | 0.0001 |
| **History of abortion** | | | |
| Yes | 27 (30.7%) | 61 (69.3%) | 0.55 |
| No | 101 (34.1%) | 195 (65.9%) | |
| **Age of the youngest child (n = 297)** | | | |
| ≤2 years | 35 (25.9%) | 100 (74.1%) | 0.04 |
| >2 year | 85 (52.5%) | 77 (47.5%) | |
| **Future plan of fertility** | | | |
| To space birth | 66 (24.2%) | 207 (75.8%) | |
| To stop birth | 62 (55.9%) | 49 (44.1%) | 0.0001 |

substantial proportion of the cases stated reasons for their IUCD use was because it was chosen by the service provider (33.6%) and husband/partner approval (23.4%).

Concerning myths and misconceptions towards the use of IUCD; 8.6% of the cases and 10.1% of controls perceive that IUCD causes infertility, 11.7% of the cases and 31.4% of the controls perceive that IUCD causes infection in the uterus, 4.7% of the cases and 31.4% of the controls perceive that IUCD causes cancer, 5.5% of the cases and 15.9% of the controls perceive that IUCD affects sexual intercourse, and 10.9% of the cases and 20.9% of the controls perceive that IUCD migrates to other body organs.

Those women who were currently using OCPs, implants or injectable (controls) were requested to indicate the reasons for not using IUCD. More than a quarter (29.4%) of the non-IUCD user's

**Table 3. Knowledge and attitude towards IUCD use and its association among reproductive age women in Addis Ababa, 2017.**

| Variables | Case (n = 128) | Control (n = 256) | *P*-value |
|---|---|---|---|
| **Number of modern contraceptive methods mentioned by the women** | | | |
| One | 5 (13.9%) | 31 (86.1%) | |
| Two | 36 (35.3%) | 66 (64.7%) | 0.004 |
| Three | 29 (27.6%) | 76 (72.4%) | |
| Four and more | 58 (41.1%) | 83 (58.9%) | |
| **Source of information about the contraceptive methods** | | | |
| Health facility (healthcare provider) | 50 (19.7%) | 204 (80.3%) | |
| Media or friends | 78 (60.0%) | 52 (40.0%) | 0.0001 |
| **Husband's/partner's opinion towards IUCD** | | | |
| Against | 29 (12.3%) | 207 (87.7%) | |
| Supporting | 99 (66.9%) | 49 (33.1%) | 0.0001 |
| **Comfortable of exposure during IUCD insertion** | | | |
| No | 58 (24.1%) | 183 (75.9%) | |
| Yes | 70 (48.9%) | 73 (51.1%) | 0.0001 |

reason for not using IUCD was fear that IUCD causes pain in pelvic region after insertion. More-over, irregular bleeding during menstruation, risk of uterine perforation, risk of cancer, not suitable with laborious work, wanted short-term methods only, husband's disapproval and increased risk of PID (infection) were also reported as reasons given by controls for not using IUCD.

In the bivariate analysis, IUCD use was associated with knowing four or more modern con-traceptive methods, information sources being media or friends, being comfortable of expos-ing private organs during insertion, supporting opinion of husband/partner, perceiving that IUCD causes infection, perceiving that IUCD affects sexual intercourse and perceiving that IUCD may migrate to other organs (Table 4).

## Determinants of IUCD use

After controlling for confounding variables in the multi-variable binary logistic regression analysis using backward stepwise method (Table 5), IUCD use was associated with hus-bands/partners being supportive, educational status, women having a perception of IUCD does not cause infection and the source of information about IUCD being mass media/friends.

Women whose husbands/partners were supportive of IUCD use were about thirteen times more likely to use IUCD than those women whose husbands/partners were against IUCD use (Adjusted OR = 13.24, 95% CI; 5.30–33.02). The odds of IUCD use among the literate women were five times higher than the illiterate women (Adjusted OR = 5.31, 95% CI; 1.05–26.93). Likewise, women who had a perception that IUCD does not cause infection were four times more likely to use IUCD than their counterparts (Adjusted OR = 4.38, 95% CI; 1.45–13.26). Similarly, the odds of IUCD use among women who received information about IUCD from mass media or friends were about four times higher as compared to those who received information from healthcare providers (Adjusted OR = 3.81, 95% CI; 1.49–9.74).

**Table 4. Perception towards IUCD use and its association among reproductive age women in Addis Ababa, 2017.**

| Variables | Case | Control | *P*-value |
|---|---|---|---|
| **Perceives that IUCD causes infertility** | | | |
| Yes | 11 (29.7%) | 26 (70.3%) | |
| No | 117 (33.7%) | 230 (66.3%) | 0.63 |
| **Perceives that IUCD causes infection** | | | |
| Yes | 15 (16.0%) | 79 (84.0%) | |
| No | 113 (49.1%) | 177 (50.9%) | 0.0001 |
| **Perceives that IUCD causes cancer** | | | |
| Yes | 6 (25.0%) | 18 (75.0%) | |
| No | 122 (33.9%) | 238 (66.1%) | 0.37 |
| **Perceives that IUCD affects sexual intercourse** | | | |
| Yes | 7 (14.9%) | 40 (85.1%) | |
| No | 121 (35.9%) | 216 (64.1%) | 0.006 |
| **Perceives that IUCD may results in pregnancy** | | | |
| Yes | 16 (59.6%) | 38 (40.4%) | 0.53 |
| No | 112 (33.9%) | 218 (66.1%) | |
| **Perceives that IUCD may migrate to other organ** | | | |
| Yes | 14 (21.2%) | 52 (78.8%) | 0.02 |
| No | 114 (35.8%) | 204 (64.2%) | |

**Table 5. Multi-variable analysis showing factors associated with IUCD use among reproductive age women in Addis Ababa, Ethiopia in 2017.**

| Variables | Cases | Control | Crude OR (95% CI) | Adjusted OR (95% CI) | P-value |
|---|---|---|---|---|---|
| **Religion** | | | | | |
| Muslim | 102 (72.9%) | 38 (27.1%) | 1.00 | 1.00 | |
| Christian | 26 (10.7%) | 218 (89.3%) | 0.68 (0.39–1.18) | 0.35 (0.10–1.25) | 0.11 |
| **Educational status** | | | | | |
| Illiterate | 7 (17.1%) | 34 (82.9%) | 1.00 | 1.00 | |
| Literate | 121 (35.3%) | 222 (64.7%) | 2.65 (1.14–6.15) | 5.31 (1.05–26.93) | 0.044 |
| **Age of the youngest child** | | | | | |
| ≤2 years | 35 (25.9%) | 100 (74.1%) | 1.00 | 1.00 | 0.066 |
| >2 year | 85 (52.5%) | 77 (47.5%) | 3.15 (0.93–5.17) | 3.24 (0.74–7.48) | |
| **Source of information** | | | | | |
| Healthcare provider | 50 (19.7%) | 204 (80.3%) | 1.00 | 1.00 | |
| Media or friends | 78 (60.0%) | 52 (40.0%) | 6.12 (3.83–9.77) | 3.81 (1.49–9.74) | 0.005 |
| **Husband/partner's opinion** | | | 1.00 | | |
| Against | 29 (12.3%) | 207 (87.7%) | 14.42 (8.59 – 24.21) | 13.24 (5.30 – 33.02) | 0.0001 |
| Supporting | 99 (66.9%) | 49 (33.1%) | | | |
| **IUCD causes infection** | | | | | |
| Yes | 15 (16.0%) | 79 (84.0%) | 1.00 | 1.00 | 0.0001 |
| No | 113 (49.1%) | 177 (50.9%) | 4.89 (2.63–9.10) | 4.38 (1.45–13.26) | |
| **IUCD may migrate** | | | | | |
| Yes | 14 (21.2%) | 52 (78.8%) | 1.00 | 1.00 | 0.085 |
| No | 114 (35.8%) | 204 (64.2%) | 3.42 (1.78–6.57) | 3.69 (0.83–16.39) | |

## Discussion

The study assessed the factors affecting the use of IUCD in Addis Ababa, in order to identify factors that may be inhibiting utilization of IUCD and to present recommendations passed on. After controlling for confounding variables; IUCD use was associated with husbands/partners being supportive, literate status, women having a perception of IUCD does not cause infection, and the source of information about IUCD being mass media/friends. However, religion and having a perception that IUCD will migrate to other organs of the body were not significantly associated.

It is interesting to note that husband's/partner's support is a strong determining factor for IUCD use in Addis Ababa. Women whose husband/partner is supportive of IUCD use were about thirteen times more likely to use IUCD than those women whose husband/partner is against IUCD use. This finding of the study is consistent with the findings from other parts of Ethiopia [16, 17, 19], Nepal [25] and Malawi [26]. Husband's/partner's support should be targeted to increase utilization of IUCDs. Particularly, in developing countries like Ethiopia, where decision making ability of the women is lower and the tradition of the country is suppressive of the women, it is very much crucial to involve husbands/partners in the information, education and communication programs to enhance utilization of IUCD [21, 27].

This study revealed that literate status was found to be significantly associated with IUCD use. This finding of the current study is consistent with those studies conducted in Ethiopia and other parts of Africa [17, 19, 20, 26, 28]. For instance, according to the finding of Ethiopian Health and Demographic Survey of 2016, the IUCD use was 0.5 percent for women with no education compared with 7 percent of women with more than secondary education [3]. This can be explained by those women with higher educational level are more likely to hold positive attitude towards IUCD use, have a greater knowledge about family planning options

and have more autonomy within families about the decision to use contraceptive methods than women with no education [25].

Long standing beliefs, myths and misconception were cited in different studies as the major factors responsible for the underuse of IUCD [25, 28]. In a similar manner, this study also revealed that perception of the women that IUCD cause infection was found to be prohibiting factor for IUCD use. Moreover, majority of the controls had perception that IUCD can cause infertility, can cause cancer, affect sexual intercourse and migrate to other organs. This finding of the current study implies that there is still lack of awareness towards IUCD among the reproductive age women in Addis Ababa.

Although all the participants of the present study heard about at least one modern family planning methods, the source of information was one of the determinants for their use of IUCD. Women who heard about IUCD from mass-media and/or friends were more likely to use IUCD than those women who heard from healthcare providers. This finding of the study calls for provision of due emphasis on the mass-media utilization to deliver important health messages related to IUCD [9, 17, 25, 29]. Mass media outlets are accepted as a reliable sources of information than other sources because of many reasons such as; mass media messages are broadcasted for millions of audiences at once, messages can be nationally monitored, the approaches to deliver mass-media messages are often explicit and in a demonstrable fashion and the mass media messages can impact other important members of the family, in this instance, husbands, whom this study demonstrates are important in choosing or refusing IUCD use [30, 31]. On the other hand, messages that are communicated by the healthcare providers working in the health centers might not be considered as reliable as mass media messages. This is because health centers are among the lowest units of health care delivery units in Ethiopia, where health officers or clinical nurses are providing services [3, 23]. This finding has implications that the quality and contents of the messages delivered by the health-care providers in the health centers might not address the information demand of the community. Therefore, the finding calls for addressing the gap through proper training on the preparation and delivery of health messages [26].

This study was conducted in multiple study areas of the city which maximized the scope of the study and helped to identify the current factors that are associated with IUCD use. However, in this study, private facilities and hospitals were not included; this might limit the generalization of the study. However, more affluent and better educated women have the means (e.g., money, knowledge, educational status, transportation) to attend private clinics and, based upon this study, are more likely to use IUCDs than the general population. Moreover, this study is solely in the urban setting and cannot be generalized for the rural women. Understanding socio-cultural determinants of IUCD use was not possible in this study, as the study was purely quantitative. Further qualitative or mixed approach studies exploring the socio-cultural determinants of IUCD use in urban and rural population of Ethiopia should be conducted. This study might be also limited by social-desirability bias with regard to reporting the myths and miss-conceptions towards IUCD and the literacy level of the participants. Absence of matched controls (e.g., in terms of age) might have also affected the finding of the study. Furthermore, multi-disciplinary studies including the social sciences perspective would be helpful to determine the role of husband's involvement in decision making process of contraceptive use.

## Conclusions

For the fertility rates continue to decline, unintended pregnancies, unsafe abortions and associated mortalities to be reduced the use of contraceptives plays an important role. However, IUCD use is determined by husbands/partners being supportive, literate status, women having

a perception of IUCD does not cause infection, the source of information about IUCD being mass media/friends and women with age of youngest child >2 years. Husband's/partner's support is of critical importance because of its strong association with IUCD use. Therefore, all the interventions to enhance utilization of IUCD should involve husband's/partner's. Moreover, due attention should also be provided for delivering IUCD related messages in the public mass-media.

## Supporting information

**S1 File. Research Ethics Committee approval letter.**
(PDF)

**S1 Appendix. Questionnaire.**
(DOCX)

**S1 Dataset. Dataset used in the study.**
(CSV)

**S2 Dataset.**
(CSV)

**S1 Checklist. Checklist of items that should be included in reports of observational studies.**
(DOCX)

## Acknowledgments

The authors are grateful to the participants of the study and data collectors for their contributions.

## Author Contributions

**Conceptualization:** Biruk Engida.

**Data curation:** Nebiyu Dereje, Biruk Engida.

**Formal analysis:** Nebiyu Dereje, Biruk Engida.

**Funding acquisition:** Roger P. Holland.

**Investigation:** Nebiyu Dereje, Biruk Engida.

**Methodology:** Nebiyu Dereje, Biruk Engida, Roger P. Holland.

**Software:** Nebiyu Dereje, Biruk Engida.

**Supervision:** Nebiyu Dereje, Roger P. Holland.

**Validation:** Nebiyu Dereje, Roger P. Holland.

**Visualization:** Nebiyu Dereje, Roger P. Holland.

**Writing – original draft:** Nebiyu Dereje.

**Writing – review & editing:** Nebiyu Dereje, Roger P. Holland.

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
