## [Decision Letter · Decision Letter 0]

1 Nov 2019

PONE-D-19-22523

Factors Associated with Intrauterine Contraceptive Device Use among Women of Reproductive Age Group in Addis Ababa, Ethiopia: A Case Control Study

PLOS ONE

Dear Mr. Dereje,

Thank you for submitting your manuscript to PLOS ONE. After careful consideration, we feel that it has merit but does not fully meet PLOS ONE’s publication criteria as it currently stands. Therefore, we invite you to submit a revised version of the manuscript that addresses the points raised during the review process.

This manuscript requires substantial revision and I ask that the authors consider the points raised by Reviewer 2 and the additional feedback in my comments in the review.  Please also provide a linked source for the data used for the study analysis within the revised manuscript.

We would appreciate receiving your revised manuscript by Dec 16 2019 11:59PM. To enhance the reproducibility of your results, we recommend that if applicable you deposit your laboratory protocols in protocols.io, where a protocol can be assigned its own identifier (DOI) such that it can be cited independently in the future. For instructions see: http://journals.plos.org/plosone/s/submission-guidelines#loc-laboratory-protocols

We look forward to receiving your revised manuscript.

Kind regards,

Catherine S. Todd

Academic Editor

PLOS ONE

Additional Editor Comments:

In addition to points raised by the reviewers, I would suggest that the authors re-structure the Discussion to discuss topics in order based on strength of association. Partner support was the strongest determinant of selecting an IUCD but is only discussed midway through this section. The limitations section also do not encompass the full range of limitations, included reported nature of many of the co-variates, the non-representative sample selection and use of non-matched controls that further limit generalizability; please make this section more robust. The manuscript would benefit from review by a native English speaker for editing and flow and the Methods section can be made more concise.

Please explain why was written consent was not obtained, how you recorded/documented participant consent, and if the ethics committees/IRBs approved this consent procedure.

3. Please include additional information regarding the survey or questionnaire used in the study and ensure that you have provided sufficient details that others could replicate the analyses. For instance, if you developed a questionnaire as part of this study and it is not under a copyright more restrictive than CC-BY, please include a copy, in both the original language and English, as Supporting Information.  If the original language is written in non-Latin characters, for example Amharic, Chinese, or Korean, please use a file format that ensures these characters are visible.

4. Please state whether you validated the questionnaire prior to testing on study participants. Please provide details regarding the validation group within the methods section.

5. Please clarify in your Methods section whether the XYZ questionnaire is published under a CC-BY license, or whether you obtained permission from the publisher to reproduce the questionnaire in this manuscript. Please explain any copyright or restrictions on this questionnaire.

6. We note you have included a table to which you do not refer in the text of your manuscript. Please ensure that you refer to Table 4 in your text; if accepted, production will need this reference to link the reader to the Table.

7. Please include captions for your Supporting Information files at the end of your manuscript, and update any in-text citations to match accordingly. Please see our Supporting Information guidelines for more information: http://journals.plos.org/plosone/s/supporting-information

Reviewers' comments:

Reviewer's Responses to Questions

**Comments to the Author**

1. Is the manuscript technically sound, and do the data support the conclusions?

Reviewer #1: Partly

Reviewer #2: Yes

2. Has the statistical analysis been performed appropriately and rigorously? 

Reviewer #1: Yes

Reviewer #2: No

3. Have the authors made all data underlying the findings in their manuscript fully available?

Reviewer #1: Yes

Reviewer #2: Yes

4. Is the manuscript presented in an intelligible fashion and written in standard English?

Reviewer #1: Yes

Reviewer #2: No

5. Review Comments to the Author

Reviewer #1: An observation study of 128 cases and 256 controls were selected at a 1:2 ratio to identify factors affecting the use of Intrauterine Contraceptive Devices (IUCDs) among women of reproductive age group in Ethiopia. Logistic regression analysis revealed that literate status, source of information about IUCD being mass media/friends, supportive partners, and women having a perception of IUCD does not cause infection were independent predictive factors of IUCD use when controlling for confounders.

Minor revisions:

1- Abstract: No need to state the sample sizes twice in the abstract.

2- Modify the results and/or conclusion to indicate that the significant factors in the multivariate logistic model were INDEPENDENT predictors of IUCD use.

3- Line 121: Provide additional details for justifying the study’s target sample size. The power calculation should include: sample size, alpha level (indicating one or two-sided), minimal detectable difference and statistical testing method.

4- Lines 173 and 185: The standard terminology for bivariate is univariate logistic regression.

5- Tables: Define all abbreviations and provide only the overall p-values rather than pairwise p-values for variables with more than two categories.

6- Tables: Include percentages corresponding to the frequencies.

7- When discussing the results of the multivariate model, clarify that four factors were significant when adjusted for the nonsignificant factors that were also included in the model. This has been stated properly in the discussion, but should also be included in the results section of the abstract.

Reviewer #2: Reviewer: Mengistu Meskele

Date: 05 October 2019

Title: Factors Associated with Intrauterine Contraceptive Device Use among Women of Reproductive Age Group in Addis Ababa, Ethiopia: A Case Control Study

Manuscript number: PONE-D-19-22523

General comments

Generally, this research title currently a public health important in Ethiopia. Though, it needs further improvement in order to adhere the publication criteria of the journal. My comments are presented below

The study presents the results of original research= Yes

2. Results reported have not been published elsewhere=Yes

3. Experiments, statistics, and other analyses are performed to a high technical standard and are described in sufficient detail. No

4. Conclusions are presented in an appropriate fashion and are supported by the data. No

5. The article is presented in an intelligible fashion and is written in Standard English. No, some improvement is needed.

6. The research meets all applicable standards for the ethics of experimentation and research integrity. Not separately explained in the method section, not ethics approval letter attached

7. The article adheres to appropriate reporting guidelines and community standards for data availability =Yes

Specific comments

Title and sub-titles should be sentence case and follow journal style of formatting manuscript

The formatting styles of tables in this manuscript are not suitable for publication. Each variable should be written in cell or should follow the journal style. Also one table should not be presented two to three pages. If possible, it would be each could be table to present in one page?

Table 4. COR and AOR should be presented along with 95% CI in one cell?

In table 1,2 and 3 the COR and 95% CI columns should be removed?

Abstract: should not be more than 300 words?

Why the definition of cases was only selected from new users of family planning? On other hand your controls were selected both from new and return clients? Definition and selection criteria for cases and controls should be the same. What is your justification for this difference?

From line 123, From all the health centers in Addis Ababa, 20 health centers (two from each sub-city) were selected by employing simple random sampling technique?

Question: How many health centers were there in AA? Why only 20 were selected?

Unnecessary capitalization and inconsistency in use of abbreviations are identified. For instance, intrauterine Device/IUCD. Sometimes you use IUCD but other time intrauterine Device. If once you abbreviated it in the first time use, use IUCD throughout?

Did you check the model fitness test? That means post estimation test? How did you controlled over matching? What about multicolloniarity?

Letter for Ethical clearance file should be attached as supporting information?

List of abbreviations should be removed from line numbers 324-331?

Discussion: You didn’t consider negative finding in your discussion. You discussed only positive finding. If possible consider it.

Strengths and limitations should be stated at end of discussions?

Presentation of 95% CI: You put it not consistently in your document throughout, sometimes you used one digit, and other times 2 digits. Be consistent and I recommend 2 digits. It is usually preferable and be consistent?

Line 236: You used p-value <0.20 in the bivariate analysis. Why p- value < 0.2? Why not 0.25?. If you could have used backward conditional stepwise regression rather than enter method, your multivariable analysis will be very sound, also your table also cannot three to four pages. Consider this?

References are not completed and an acceptable label.

References 2, 3,8,9,10,11,16,17,18,22,23,28,29,30,31, are either not completed or URL and accessed dates are missed, also the authors naming are not correct. List of more than 6 authors, you should write the first 6 authors and use et al.?

Financial disclosure, conflict of interest and source of funding are not mentioned.

Conclusion

In the conclusion section, the statement “For the fertility rates continue to decline, unintended pregnancies, unsafe abortions and 317 associated mortalities to be reduced the use of contraceptives plays an important role”. This sentence can be removed and the author can bring conclusions from his/her finding rather than stating the above unrelated issues of the study.

6. PLOS authors have the option to publish the peer review history of their article (what does this mean?). If published, this will include your full peer review and any attached files.

Reviewer #1: No

Reviewer #2: Yes: Mengistu Meskele

---

## [Author Response · Author response to Decision Letter 0]

26 Nov 2019

Point by point response

PONE-D-19-22523

Factors Associated with Intrauterine Contraceptive Device Use among Women of Reproductive Age Group in Addis Ababa, Ethiopia: A Case Control Study

Dear editor and reviewers, 

Thank you for your comments and suggested revisions that have helped us to improve the manuscript. We have revised our manuscript according to your constructive and very insightful comments and suggestions. Attached you will find our revised manuscript and below you will find a response to each of your comments.

Response for editor’s comment

In addition to points raised by the reviewers, I would suggest that the authors re-structure the Discussion to discuss topics in order based on strength of association. Partner support was the strongest determinant of selecting an IUCD but is only discussed midway through this section. The limitations section also do not encompass the full range of limitations, included reported nature of many of the co-variates, the non-representative sample selection and use of non-matched controls that further limit generalizability; please make this section more robust. The manuscript would benefit from review by a native English speaker for editing and flow and the Methods section can be made more concise.

Response: Thank you for the suggestion. The discussion is re-structured now to discuss the strongest predictor first. Also, limitations of the study is now detailed. The revised manuscript is also reviewed by native English speaker.

Response for reviewer 2 comments and suggestions

R2: Title and sub-titles should be sentence case and follow journal style of formatting manuscript

Response: Thank you, now it is corrected.

R2: The formatting styles of tables in this manuscript are not suitable for publication. Each variable should be written in cell or should follow the journal style. Also one table should not be presented two to three pages. If possible, it would be each could be table to present in one page?

Response: Thank you for the comment. The formatting of the tables are now corrected.

R2: Table 4. COR and AOR should be presented along with 95% CI in one cell?

Response: Thank you, COR and AOR are now presented along with 95% CI in one cell.

R2: In table 1,2 and 3 the COR and 95% CI columns should be removed?

Response: Thank you for the suggestion. COR and 95% CI are removed from the tables. 

R2: Abstract: should not be more than 300 words?

Response: Thank you, it is corrected.

R2: Why the definition of cases was only selected from new users of family planning? On other hand your controls were selected both from new and return clients? Definition and selection criteria for cases and controls should be the same. What is your justification for this difference?

From line 123, From all the health centers in Addis Ababa, 20 health centers (two from each sub-city) were selected by employing simple random sampling technique?

Response: Thank you for the comment. Actually, we have included only new cases and controls to reduce the potential biases. Now, this is corrected throughout the revised manuscript. 

Twenty health centers out of the total 98 health centers were selected by simple random sampling technique (lottery method).

R2: Question: How many health centers were there in AA? Why only 20 were selected?

Response: There were 98 health centers in Addis Ababa. To increase the scope and ensure representativeness of the population attending care, we randomly selected 20 health centers (two from each sub-city). It was not feasible to take more than 20 health centers. We employed simple random sampling technique to ensure equal chance of being included in the study.

R2: Unnecessary capitalization and inconsistency in use of abbreviations are identified. For instance, intrauterine Device/IUCD. Sometimes you use IUCD but other time intrauterine Device. If once you abbreviated it in the first time use, use IUCD throughout?

Response: Thank you. IUCD is used throughout the manuscript.

R2: Did you check the model fitness test? That means post estimation test? How did you controlled over matching? What about multicolloniarity?

Response: Yes, we have checked model fitness test by Hosmer-Lemeshow goodness-of-fit test. Also, we have checked multicoliniarity by using the multicoliniarity diagnostics (VIF and Tolerance test).

R2: Letter for Ethical clearance file should be attached as supporting information?

Response: Attached as supporting information

R2: List of abbreviations should be removed from line numbers 324-331?

Response: Thank you, addressed.

R2: Discussion: You didn’t consider negative finding in your discussion. You discussed only positive finding. If possible consider it.

Response: Thank you. We have now considered discussing pertinent negative findings in the discussion.

R2: Strengths and limitations should be stated at end of discussions?

Response: We have detailed the strengths and limitations of the study

R2: Presentation of 95% CI: You put it not consistently in your document throughout, sometimes you used one digit, and other times 2 digits. Be consistent and I recommend 2 digits. It is usually preferable and be consistent?

Response: Thank you for the comment. We have now put the numbers consistently

R2: Line 236: You used p-value <0.20 in the bivariate analysis. Why p- value < 0.2? Why not 0.25?. If you could have used backward conditional stepwise regression rather than enter method, your multivariable analysis will be very sound, also your table also cannot three to four pages. Consider this?

Response: Thank you for the comment. We used P value <0.25 as a criteria to enter variables into the multi-variable model. Also, we have now re-analyzed our data using the backward conditional stepwise binary logistic regression model and the findings are reported.

R2: References are not completed and an acceptable label.

References 2, 3,8,9,10,11,16,17,18,22,23,28,29,30,31, are either not completed or URL and accessed dates are missed, also the authors naming are not correct. List of more than 6 authors, you should write the first 6 authors and use et al.?

Response: Thank you, the formatting of the references is now corrected.

R2: Financial disclosure, conflict of interest and source of funding are not mentioned.

Response: Based on the PLOS ONE submission guidelines, statements on financial disclosure and conflict of interest were given in the submission form in the system.

R2: In the conclusion section, the statement “For the fertility rates continue to decline, unintended pregnancies, unsafe abortions and 317 associated mortalities to be reduced the use of contraceptives plays an important role”. This sentence can be removed and the author can bring conclusions from his/her finding rather than stating the above unrelated issues of the study.

Response: The introductory sentence of the conclusion is used to link our findings with its public health importance. Then, we have concluded the study based on the findings of the study.

---

## [Decision Letter · Decision Letter 1]

7 Jan 2020

PONE-D-19-22523R1

Factors associated with intrauterine contraceptive device use among women of reproductive age group in Addis Ababa, Ethiopia: A case control study

PLOS ONE

Dear Mr. Dereje,

Thank you for submitting your manuscript to PLOS ONE. After careful consideration, we feel that it has merit but does not fully meet PLOS ONE’s publication criteria as it currently stands. Therefore, we invite you to submit a revised version of the manuscript that addresses the points raised during the review process.

We would appreciate receiving your revised manuscript by Feb 21 2020 11:59PM. To enhance the reproducibility of your results, we recommend that if applicable you deposit your laboratory protocols in protocols.io, where a protocol can be assigned its own identifier (DOI) such that it can be cited independently in the future. For instructions see: http://journals.plos.org/plosone/s/submission-guidelines#loc-laboratory-protocols

We look forward to receiving your revised manuscript.

Kind regards,

Catherine S. Todd

Academic Editor

PLOS ONE

Additional Editor Comments (if provided):

The authors have been largely responsive to previous comments but both reviewers and I note that the authors disregarded responding to several comments in the first review. There was no response to Reviewer 1's critiques; Reviewer 2 noted several comments that did not receive a response, and I note that the Limitations section still does not include some major limitations that were brought to the authors' attention in the first review, namely the reported nature of several of the covariates (e.g., literacy) and the absence of matched controls. Further, though you state the revised manuscript was reviewed by a native English speaker, there remain multiple places where verb tense, punctuation, and other grammatical errors are present or examples of awkward phrasing (which can be adjusted and reduce word count in the abstract). For example: "In each health center, all the eligible cases were included in the study consecutively until the desired sample size was achieved. For each case, two adjacent controls were selected to be included in the study." can be improved to "In each selected health center, all eligible cases were enrolled consecutively until sample size was achieved. Two consecutive controls were selected for each case." Last, there appear to be different fonts within the manuscript and portions do not comply with formatting requirements. Please attend to all of these matters completely and in the response letter, please state what was done in response and provide illustrative text rather than stating the query was addressed, as this was not consistently the case with the last revision. Thus, you are invited to revise and re-submit.

Reviewers' comments:

Reviewer's Responses to Questions

**Comments to the Author**

1. If the authors have adequately addressed your comments raised in a previous round of review and you feel that this manuscript is now acceptable for publication, you may indicate that here to bypass the “Comments to the Author” section, enter your conflict of interest statement in the “Confidential to Editor” section, and submit your "Accept" recommendation.

Reviewer #1: (No Response)

Reviewer #2: All comments have been addressed

2. Is the manuscript technically sound, and do the data support the conclusions?

Reviewer #1: Yes

Reviewer #2: Yes

3. Has the statistical analysis been performed appropriately and rigorously? 

Reviewer #1: Yes

Reviewer #2: Yes

4. Have the authors made all data underlying the findings in their manuscript fully available?

Reviewer #1: Yes

Reviewer #2: Yes

5. Is the manuscript presented in an intelligible fashion and written in standard English?

Reviewer #1: Yes

Reviewer #2: Yes

6. Review Comments to the Author

Reviewer #1: Minor revisions:

1- Abstract: No need to state the sample sizes twice in the abstract.

2- Modify the results and/or conclusion to indicate that the significant factors in the multivariate logistic model were INDEPENDENT predictors of IUCD use.

3- Line 121: Provide additional details for justifying the study’s target sample size. The power calculation should include: sample size, alpha level (indicating one or two-sided), minimal detectable difference and statistical testing method.

4- Lines 173 and 185: The standard terminology for bivariate is univariate logistic regression.

5- Tables: Define all abbreviations and provide only the overall p-values rather than pairwise p-values for variables with more than two categories.

6- Tables: Include percentages corresponding to the frequencies.

7- When discussing the results of the multivariate model, clarify that four factors were significant when adjusted for the nonsignificant factors that were also included in the model. This has been stated properly in the discussion, but should also be included in the results section of the abstract.

Reviewer #2: Minor comments

1. Contribution: Equal or unequal contribution should be explained at first page?

2. Style: Font style for title and headings should be adhered journal style?

3. Table 5. Cross tabulation for cases and controls should be included?

7. PLOS authors have the option to publish the peer review history of their article (what does this mean?). If published, this will include your full peer review and any attached files.

Reviewer #1: No

Reviewer #2: Yes: Mengistu Meskele

---

## [Author Response · Author response to Decision Letter 1]

13 Jan 2020

Point by point response

PONE-D-19-22523

Factors Associated with Intrauterine Contraceptive Device Use among Women of Reproductive Age Group in Addis Ababa, Ethiopia: A Case Control Study

Dear editor and reviewers, 

Thank you for your comments and suggested revisions that have helped us to improve the manuscript. We have revised our manuscript according to your constructive and very insightful comments and suggestions. Attached you will find our revised manuscript and below you will find a response to each of your comments.

Response for editor’s comment

Editor: The authors have been largely responsive to previous comments but both reviewers and I note that the authors disregarded responding to several comments in the first review. There was no response to Reviewer 1's critiques; Reviewer 2 noted several comments that did not receive a response, and I note that the Limitations section still does not include some major limitations that were brought to the authors' attention in the first review, namely the reported nature of several of the covariates (e.g., literacy) and the absence of matched controls. Further, though you state the revised manuscript was reviewed by a native English speaker, there remain multiple places where verb tense, punctuation, and other grammatical errors are present or examples of awkward phrasing (which can be adjusted and reduce word count in the abstract). For example: "In each health center, all the eligible cases were included in the study consecutively until the desired sample size was achieved. For each case, two adjacent controls were selected to be included in the study." can be improved to "In each selected health center, all eligible cases were enrolled consecutively until sample size was achieved. Two consecutive controls were selected for each case." Last, there appear to be different fonts within the manuscript and portions do not comply with formatting requirements. Please attend to all of these matters completely and in the response letter, please state what was done in response and provide illustrative text rather than stating the query was addressed, as this was not consistently the case with the last revision. Thus, you are invited to revise and re-submit.

Response: Thank you for the comments. The prior request sent to us was to review comments/suggestions based on the comments from the editor and reviewer 2. That is why we didn’t reply for the reviewer 1 questions. Now, we have replied to the reviewer 1 queries too.

The limitations of the study is also further detailed to address the reported nature of the covariates and the absence of matched controls. The study might be also limited by social-desirability bias with regard to reporting the myths and miss-conceptions towards IUCD and the literacy level of the participants. Absence of matched controls (e.g., in terms of age) might have also affected the findings of this study.

Consistency of font styles and formats are addressed now. To further improve the overall flow of the manuscript, it is reviewed by a native English speaker.

Response for Reviewer 1 comments

R1: 1- Abstract: No need to state the sample sizes twice in the abstract.

Response: Thank you, it is addressed. Sample size is mentioned once in the revised manuscript.

R1: Modify the results and/or conclusion to indicate that the significant factors in the multivariate logistic model were INDEPENDENT predictors of IUCD use.

Response: Thank you, this is indicated now in the revised manuscript. The adjusted odds ratio also indicates that the predictors are independently associated with the IUCD use after the confounding factors are controlled.

R1: Line 121: Provide additional details for justifying the study’s target sample size. The power calculation should include: sample size, alpha level (indicating one or two-sided), minimal detectable difference and statistical testing method.

Response: Thank you for the comment. It is addressed and all the entities for sample size calculations were given.

R1: Tables: Define all abbreviations and provide only the overall p-values rather than pairwise p-values for variables with more than two categories.

Response: Thank you for the suggestion. All abbreviations defined. The overall p-values were given in the tables.

R1: Tables: Include percentages corresponding to the frequencies.

Response: Thank you, percentages are now included.

R1: When discussing the results of the multivariate model, clarify that four factors were significant when adjusted for the nonsignificant factors that were also included in the model. This has been stated properly in the discussion, but should also be included in the results section of the abstract.

Response: Thank you for the comment. Statement clarifying that the associated factors were significant when adjusted for the nonsignificant factors that were also included in the model.

Response for Reviewer 2 comments

R2: Contribution: Equal or unequal contribution should be explained at first page?

Response: Thank you. It is included now. All the authors contributed equally to this work.

R2: Style: Font style for title and headings should be adhered journal style?

Response: Thank you. Font styles are adhering to the journal style.

R2: Table 5. Cross tabulation for cases and controls should be included?

Response: Thank you, it is included now.

---

## [Decision Letter · Decision Letter 2]

30 Jan 2020

Factors associated with intrauterine contraceptive device use among women of reproductive age group in Addis Ababa, Ethiopia: A case control study

PONE-D-19-22523R2

Dear Dr. Dereje,

We are pleased to inform you that your manuscript has been judged scientifically suitable for publication and will be formally accepted for publication once it complies with all outstanding technical requirements.

With kind regards,

Catherine S. Todd

Academic Editor

PLOS ONE

Additional Editor Comments (optional):

My thanks to the authors for addressing the remaining concerns from the reviewers and this manuscript is accepted. Upon acceptance, I would recommend professional editing as the manuscript would benefit from further grammatical and phrasing improvements in an effort to make the text more concise.

Reviewers' comments:

Reviewer's Responses to Questions

**Comments to the Author**

1. If the authors have adequately addressed your comments raised in a previous round of review and you feel that this manuscript is now acceptable for publication, you may indicate that here to bypass the “Comments to the Author” section, enter your conflict of interest statement in the “Confidential to Editor” section, and submit your "Accept" recommendation.

Reviewer #1: All comments have been addressed

Reviewer #2: All comments have been addressed

2. Is the manuscript technically sound, and do the data support the conclusions?

Reviewer #1: (No Response)

Reviewer #2: Yes

3. Has the statistical analysis been performed appropriately and rigorously? 

Reviewer #1: (No Response)

Reviewer #2: Yes

4. Have the authors made all data underlying the findings in their manuscript fully available?

Reviewer #1: (No Response)

Reviewer #2: Yes

5. Is the manuscript presented in an intelligible fashion and written in standard English?

Reviewer #1: (No Response)

Reviewer #2: Yes

6. Review Comments to the Author

Reviewer #1: (No Response)

Reviewer #2: (No Response)

7. PLOS authors have the option to publish the peer review history of their article (what does this mean?). If published, this will include your full peer review and any attached files.

Reviewer #1: No

Reviewer #2: Yes: Mengistu Meskele

---

## [Editor Report · Acceptance letter]

3 Feb 2020

PONE-D-19-22523R2 

Factors associated with intrauterine contraceptive device use among women of reproductive age group in Addis Ababa, Ethiopia: A case control study 

Dear Dr. Dereje:

I am pleased to inform you that your manuscript has been deemed suitable for publication in PLOS ONE. Congratulations! Your manuscript is now with our production department. 

With kind regards,

on behalf of

Dr. Catherine S. Todd 

Academic Editor

PLOS ONE